# Neuroanatomical Study and Three-Dimensional Cranial Reconstruction of the Brazilian Albian Pleurodiran Turtle *Euraxemys essweini*

Marcos Martín-Jiménez *[ID] and Adán Pérez-García [ID]

Grupo de Biología Evolutiva, Facultad de Ciencias, UNED, Avda. Esparta s/n, Las Rozas, 28232 Madrid, Spain
* Correspondence: mmartinjimenez@gmail.com

**Abstract:** Pleurodira represent one of the two clades that compose the crown Testudines, and their temporal range is Late Jurassic–present. However, knowledge about the neuroanatomy of extinct pleurodires is still very limited. In this context, scarce neuroanatomical information about the Cretaceous clade Euraxemydidae is currently available, limited to some characters of the Moroccan Cenomanian *Dirqadim schaefferi*. In the present work, we perform the detailed neuroanatomical study of its sister taxon, the Brazilian Albian *Euraxemys essweini*, based on the analysis of the skull of its holotype and only known individual of the species. The detailed virtual three-dimensional reconstruction of all its cranial bones is performed, also improving the information about its osseous anatomy. The different neuroanatomical cavities (i.e., cranial, nasal, and labyrinthic ones) and canals (i.e., nervous and circulatory ones) are compared with those identified thus far for other extinct and extant members of the Pleurodira in order to characterize the neuroanatomy of the extinct clade Euraxemydidae in detail.

**Keywords:** Pleurodira; Euraxemydidae; neuroanatomy; computed tomography; South America





## 1. Introduction

Pleurodira represent one of the lineages that compose the clade Testudines. Pan-pleurodiran turtles date to the Late Jurassic period and continue to be part of the current biodiversity [1–5]. Although they are currently restricted to the southern hemisphere, numerous pleurodiran lineages are identified in the fossil record of several Laurasian continents, showing great success in both abundance and diversity (e.g., [6–9]). The greatest diversity of the Pleurodira is recognized for the fossil record of both South America and Africa (e.g., [10–13]). In this sense, lineages exclusive to these continents, along with others which also inhabited Laurasia, are thus identified. In this context, Euraxemydidae is exclusively identified by two representatives: *Euraxemys essweini* Gaffney, Tong, and Meylan, 2006 [14] from the Albian (Santana Formation) of Brazil; and *Dirqadim schaefferi* Gaffney, Tong, and Meylan, 2006 [14] from the Cenomanian (Kem Kem Group) of Morocco. *Dirqadim schaefferi* was described from a nearly complete skull (its holotype) and an additional partial skull [14]. *Euraxemys essweini* is only known through an individual, corresponding to a nearly complete skeleton, including its almost-complete skull [14,15]. Gaffney and Meylan [15] presented an undefined new form from the Albian of the Santana Formation (subsequently proposed as the holotype of *Euraxemys essweini* by Gaffney et al. [14]), being positioned in an undefined position within Pelomedusoides. Several years later, Meylan [16] indicated that it was probably closely related to *Araripemys barretoi* (Price, 1973 [17]). Gaffney et al. [14] attributed this specimen to a new genus and species, *Euraxemys essweini*, included in a new lineage of Pelomedusoides, together with *Dirqadim schaefferi*: Euraxemydidae.

Knowledge about the neuroanatomy in Testudinata has increased remarkably in recent years thanks to the studies based on the CT methodology [18]. In addition to the analysis

of stem turtles (i.e., members of Testudinata not belonging to Testudines) [19–24] and extinct cryptodires [25,26], some extinct members of the Pleurodira have also been analyzed [12,27–30]. Many of these recent works also include information about extant cryptodiran or pleurodian representatives, with the aim of improving the comparative framework on the shape of some neuroanatomical structures, such as the otic region or the cranial cavity, in order to analyze adaptive implications (e.g., [21,31–33]). In addition, some studies are focused on the characterization of specific cranial structures, analyzing both their variation in different groups and their taxonomic significance [34,35]. Despite the remarkable increase in knowledge about the neuroanatomy in extinct taxa or lineages of Pleurodira (mainly belonging to the crown Podocnemidoidea), current information about the neuroanatomy of the euraxemydids is very scarce. Only the superficial three-dimensional model of the skull, that of the virtually isolated basisphenoid, and few neuroanatomical structures (i.e., the partial facial nerve canals, the right canalis cavernosus, the right canalis stapedio-temporalis, and the carotid canals) of *Dirqadim schaefferi* have been reconstructed (see Supplementary 3D models in [27]). In this context, the objective of this paper is to provide the first complete virtual three-dimensional neuroanatomical reconstruction and description of a representative of Euraxemydidae. It is based on the study of *Euraxemys essweini*. In addition, virtual three-dimensional reconstructions of its complete skull and of each cranial bone are performed, with new characters being recognized for this taxon but also for the first time in Euraxemydidae.

Institutional abbreviations: FR, Forschungsinstitut Senckenberg, Frankfurt (Germany).

Anatomical abbreviations: asc, anterior semicircular canal; bo, basioccipital; bs, basisphenoid; cas, canalis alveolaris superior; cc, crus communis; ccb, canalis caroticus basisphenoidalis; ccv, canalis cavernosus; cer, cerebral hemispheres; cl, cavum labyrinthicum; cnv, canalis nervus vidianus; col, columella auris; cp, clinoid process; cprv, canalis pro ramo nervi vidiani; cr, cartilaginous ridge; cst, canalis stapedio-temporalis; ct, cavum tympani; eo, exoccipital; faca, foramen anterius canalis nervi abducentis; faccb, foramen anterius canalis carotici basisphenoidalis; faf, fossa acustico-facialis; feng, foramen externum nervi glossopharyngei; fja, foramen jugulare anterius; fjp, foramen jugulare posterius; fnh, foramen nervi hypoglossi; fnt, foramen nervi trigemini; fnv, foramen nervi vidiani; fpca, foramen posterius canalis nervi abducentis; fpcci, foramen posterius canalis carotici interni; fper, fenestra perilymphatica; fpp, foramen palatinum posterius; fpr, foramen praepalatinum; fr, frontal; fst, foramen stapedio-temporale; gg, geniculate ganglion; hyo, hyomandibular branch of the facial nerve; ica, incisura columellae auris; IX, glossopharyngeal nerve; ju, jugal; lsc, lateral semicircular canal; med, medulla oblongata; mx, maxilla; nas, nasal cavity; npd, nasopharyngeal duct; olfd, olfactory duct; op, opisthotic; pa, parietal; pal, palatine; pf, prefrontal; pif, processus interfenestralis of the opisthotic; pit, pituitary fossa; pitg, pituitary gland; pm, premaxilla; po, postorbital; pr, prootic; psc, posterior semicircular canal; pt, pterygoid; ptp, processus trochlearis pterygoidei; qj, quadratojugal; qu, quadrate; scv, sulcus cavernosus; spp, sulcus palatinopterygoideus; sq, squamosal; so, supraoccipital; sot, septum orbitotemporale; V, trigeminal nerve; VI, abducens nerve; VIII, vestibulocochlear nerve; vo, vomer; vpe, ventral process of the exoccipital; X-XI, vagus and accessory nerves; XII, hypoglossal nerve.

## 2. Materials and Methods

The skull of the holotype and only known specimen of *Euraxemys essweini*, FR 4922, was scanned at the Senckenberg CT-Lab of Frankfurt/Main (lab code "SGN-SF-3D-Xray-CT") using a ProCon-X-ray-Micro-CT scanner. This scanner contains a 100 kV Finefocus tube, and the CT-Aquire was performed with the Fraunhofer Package, obtaining archives in .rek format. The parameters used were a voltage of 90 kV and a current of 89 μA, performing a scan with 2400 projections over 4 h. The images obtained were converted to DICOM data for the subsequent processing using VGStudio MAX, resulting in 2054 image files with a resolution of 22.3 μm. The segmentation of the right paired bones, the medial odd bones, and the neuroanatomical structures (i.e., cranial, nasal, and labyrinth cavities and nervous and circulatory canals) was performed manually using the software Avizo 7.1

(VSG). Due to the large size of the files, the images had to be segmented into packages of 400 images and subsequently fused using tools in the Geomagic Studio 2014.3.0 software. All the bones were reconstructed independently to observe their internal structure, with the aim of potentially providing additional anatomical data to those already known for the taxon (see [14]). The neuroanatomical structures were measured following criteria proposed in previous studies [27,29], using tools in Avizo 7.1 for the linear measurements and in Geomagic Studio 2014.3.0 for the volumetric ones. Bidimensional images were rendered using the snapshot tools of Avizo 7.1. The dorsal and ventral view photographs of the skull were taken at the Senckenberg Museum of Frankfurt by S. Tränkner. Finally, the figures were composed using the software Adobe Photoshop CS6. The neuroanatomical and new anatomical characters identified for *Euraxemys essweini* were compared with those known for other extant and extinct pleurodiran turtles.

## 3. Systematic Palaeontology

Testudines Batsch, 1788 [36]
Pleurodira Cope, 1864 [37]
Pelomedusoides Broin, 1988 [10]
Euraxemydidae Gaffney, Tong, and Meylan, 2006 [14]
*Euraxemys* Gaffney, Tong, and Meylan, 2006 [14]
*Euraxemys essweini* Gaffney, Tong, and Meylan, 2006 [14]
(Figures 1–4)

Material: FR 4922 is the holotype and, thus far, the only known specimen of the euraxemydid *Euraxemys essweini* and corresponds to a nearly complete skeleton from the Early Cretaceous (Albian) Santana Formation of the Brazilian Araripe Basin. The specimen included the well-preserved skull, which is analyzed here (Figures 1–4).

New anatomical characters for *Euraxemys essweini*: Gaffney et al. [14] provided a relatively detailed anatomical description of the specimen analyzed here. However, thanks to the analysis of the files obtained by the computed tomography scan of the skull, some bones can be more precisely characterized, and some structures are identified for the first time. The shape of the pituitary fossa can be well characterized, being oval and slightly longer than wide (with a length of 3.4 mm and a width of 2.8 mm; Figure 2A–C). The presence of both high clinoid processes and small foramina in the lateral portions of the pituitary fossa are recognized (Figure 2D). The canals of the abducens nerves (cranial nerve VI) are identified traversing the basisphenoid to a half-length of it and emerge laterally on the dorsal surface of the basisphenoid (Figure 2C). The new information allows us to rule out the presence of a foramen caroticum laterale. The proximal end of the right columella auris is preserved in its original position between the prootic and the opisthotic (Figure 3E). The columella only retains the footplate and a small portion of the stapes bar (Figure 3F,G). The footplate has a rounded section at the relatively wide articular region (Figure 3H).

Neuroanatomical description: The anterior region of the cranial cavity, corresponding to the olfactory duct, is ascendant from the nasal cavity to reach the cerebral hemispheres area, showing a convex dorsal surface (Figure 4C). The posterior area of the dorsal surface, from the cerebral hemispheres, is descendent to the medulla oblongata. On that surface, the angle between the forebrain and the hindbrain is almost 156°. The olfactory duct is narrow, and it represents over 20% of the cranial cavity length (Figure 4A). The cerebral hemispheres region is laterally expanded, defining the widest area of the cranial cavity. The ratio between the maximum width and the length of the cranial cavity is about 0.34. That width represents 1.66 times the width of the medulla oblongata. A low protuberance, corresponding to the cartilaginous ridge, can be observed postero-medially to the hemispheres (Figure 4A,C). That ridge is relatively short (i.e., representing the 11.3% of the length of the cranial cavity) and low (i.e., equivalent to 8.4% of the maximum height of that cavity). Ventrally, the oval pituitary fossa is located at the same level of the cartilaginous ridge (Figure 4B). The anteroposterior axis of the pituitary fossa is slightly larger than the medio-lateral one, representing a ratio of 1.2 between them. The medulla oblongata is relatively high, being at

73% of the maximum height of the cranial cavity (Figure 4C). The dorsal surface of that structure is concave.

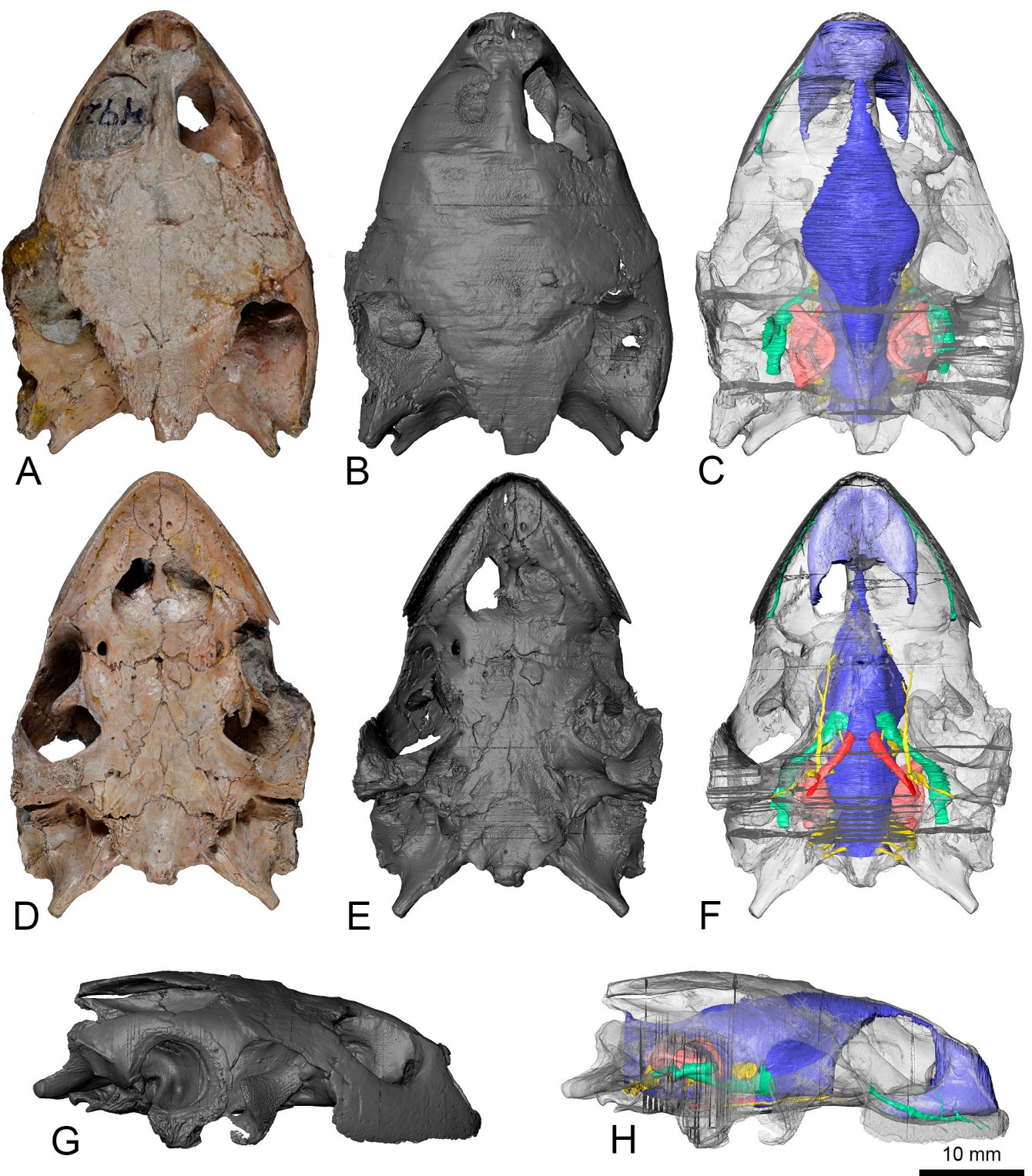

**Figure 1.** FR 4922, skull of the holotype of *Euraxemys essweini* (Euraxemydidae) from the Albian of Araripe Basin in Brazil. Photographs reproduced with permission from S. Tränkner, Senckenberg (**A**,**D**); rendered three-dimensional reconstruction (**B**,**E**,**G**); and transparent rendered skull including the three-dimensional reconstruction of the neuroanatomical structures in dorsal (**A**–**C**), ventral (**D**–**F**), and right lateral (**G**,**H**) views.

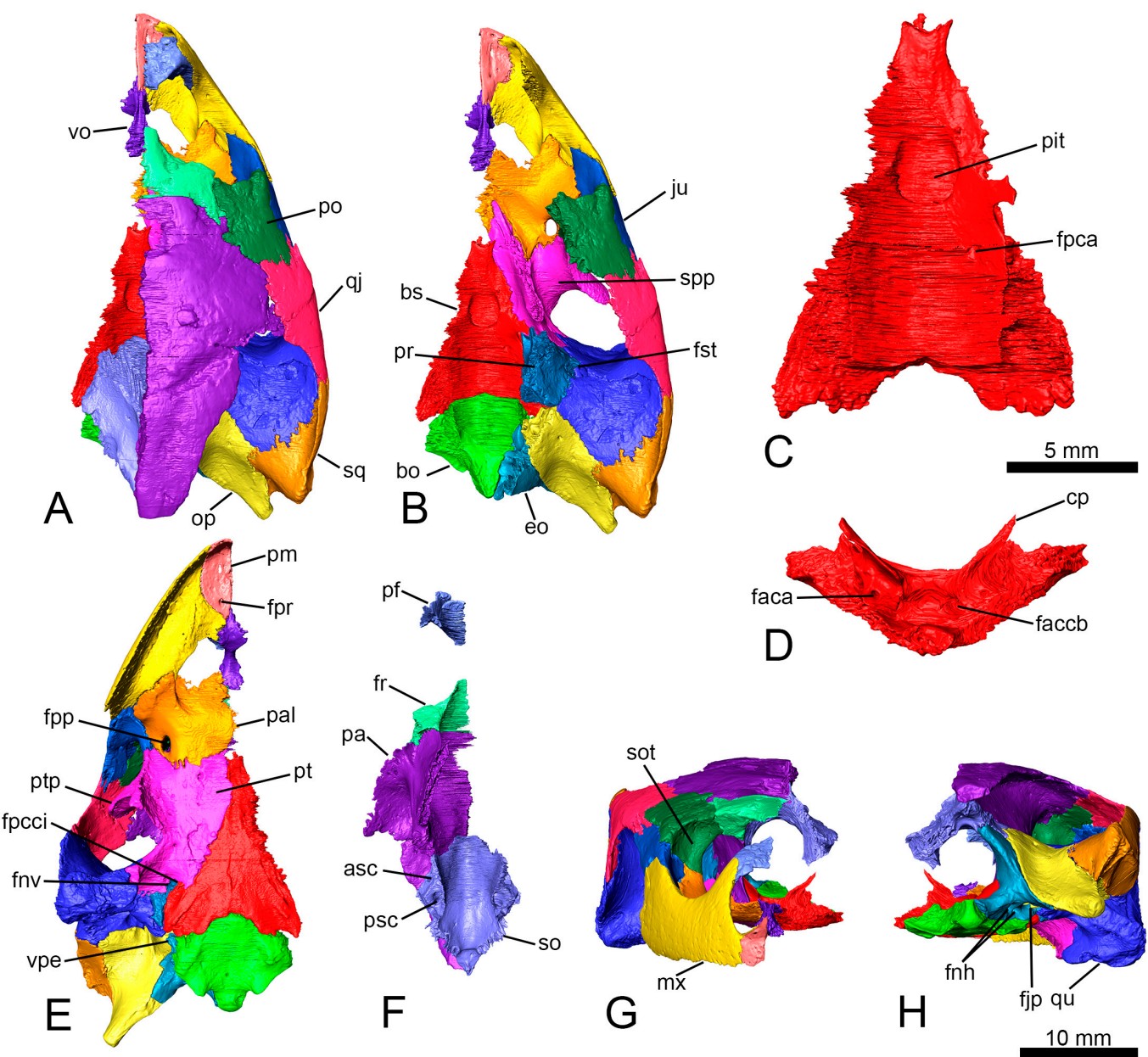

**Figure 2.** FR 4922, three-dimensional reconstruction of the skull of the holotype of *Euraxemys essweini* (Euraxemydidae) from the Albian of Araripe Basin in Brazil showing the right bones and medial odd bones virtually removing the sediment in dorsal (**A**,**B**), ventral (**E**,**F**), anterior (**G**), and posterior (**H**) views. Each bone is shown with a different color. The prefrontal, the frontal, the parietal, and the supraoccipital have been virtually removed (**B**) to show the dorsal palatal and braincase bones. The cranial roof bones are shown in ventral view (**F**). Basisphenoid (**C**,**D**) in dorsal (**C**) and anterior (**D**) views.

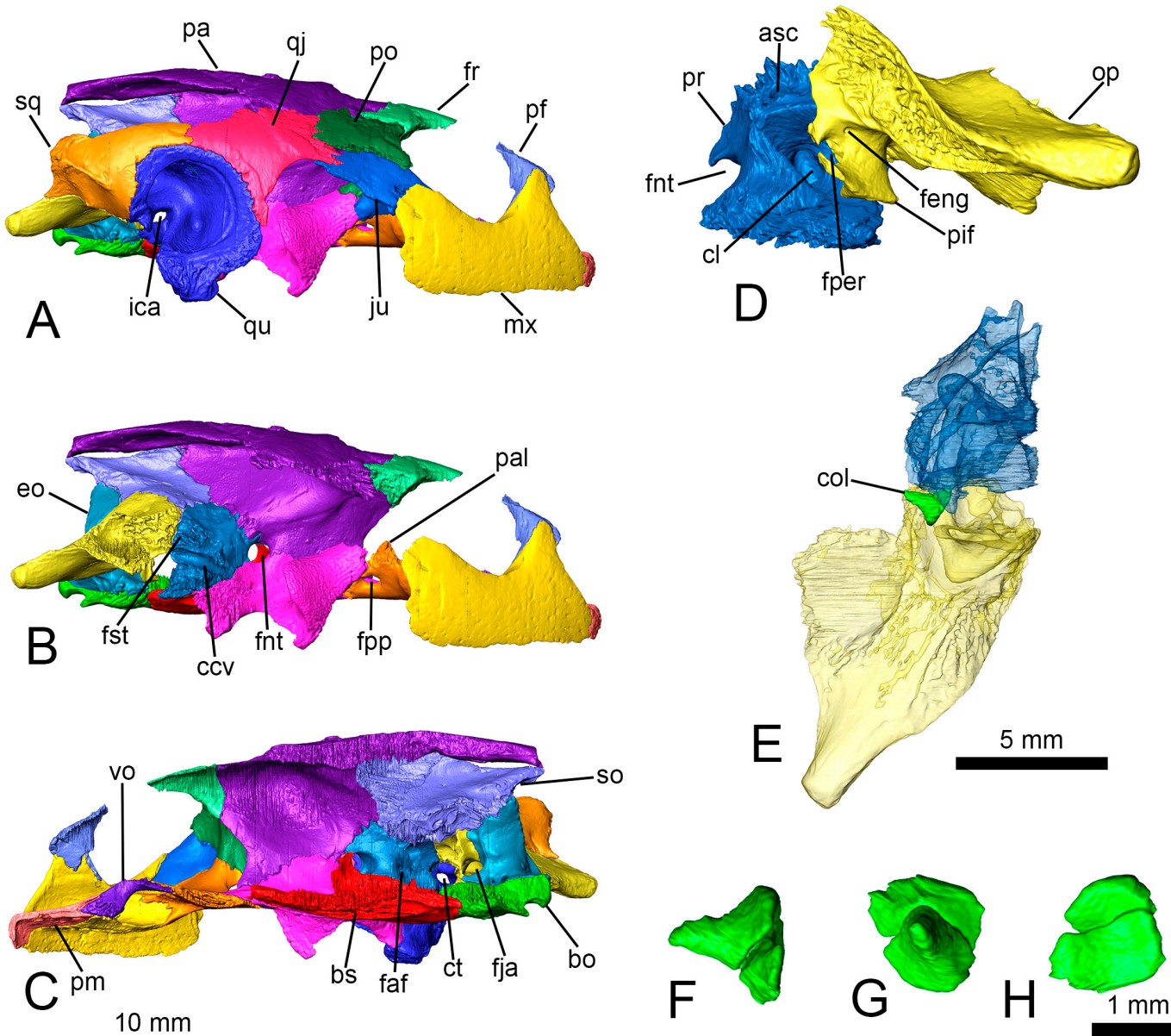

**Figure 3.** FR 4922, three-dimensional reconstruction of the skull of the holotype of *Euraxemys essweini* (Euraxemydidae) from the Albian of Araripe Basin in Brazil showing the right bones and medial odd bones virtually removing the sediment in right lateral (**A**,**B**) and medial (**C**) views. Each bone is shown with a different color. The jugal, the postorbital, the quadratojugal, the quadrate, and the squamosal have been virtually removed (**B**) to show the inner region of the temporal fossa. Prootic and opisthotic (**D**,**E**) in posterolateromedial (**D**) and ventral (**E**) views. The position of the columella auris is shown between the prootic and the opisthotic (**E**). Columella auris (**F**–**H**) in dorsal (**F**), lateral (**G**), and medial (**H**) views.

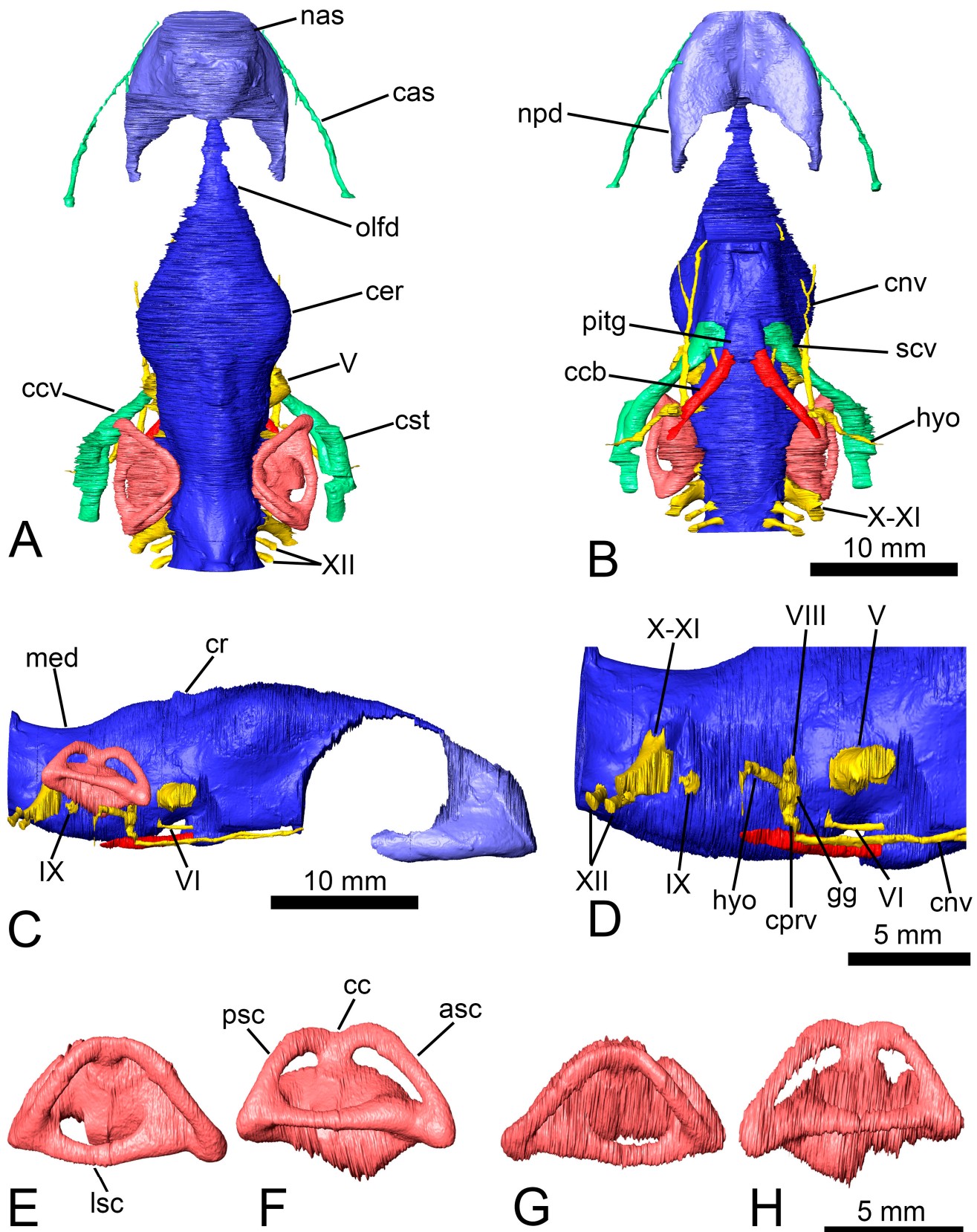

**Figure 4.** FR 4922, neuroanatomical three-dimensional virtual reconstructions of the skull of the holotype of *Euraxemys essweini* (Euraxemydidae) from the Albian of Araripe Basin in Brazil in dorsal (**A**), ventral (**B**), and lateral (**C**,**D**) views. Right (**E**,**F**) and left (**G**,**H**) inner ears in dorsal (**E**,**G**) and lateral (**F**–**H**) views.

The trigeminal nerve (cranial nerve V) exited from the cranial cavity through the foramen nervi trigemini (Figure 4A–D). This foramen is formed by the parietal anterodorsally, the prootic posterodorsally, and the pterygoid ventrally (Figure 3B). The canalis cavernosus runs posterolaterally below the foramen nervi trigemini (Figure 4A,B). The anterior region of the canalis cavernosus reaches the sulcus cavernosus (Figure 4B), being located lateral to the pituitary fossa. The posterior end of the canalis cavernosus contacts the canalis stapedio-temporalis, which presents an anterodorsal trajectory to reach the foramen stapedio-temporale (Figure 4A). This foramen is surrounded by the quadrate laterally and the prootic medially (Figure 2B). The abducens nerve pierced the basisphenoid through a straight canal (Figure 4B–D). The posterior foramen of the canal is located on the dorsal surface of the basisphenoid, posterolateral to the pituitary fossa (Figure 2C). The anterior foramen contacts the sulcus cavernosus laterally to the mid-length of the fossa (Figure 2D). The facial and vestibulocochlear nerves (cranial nerves VII and VIII) left the cranial cavity through two foramina housed in the fossa acustico-facialis of the prootic (Figures 3C and 4B–D). The vestibulocochlear nerve crossed a short canal and innervated the cavum labyrinthicum. The facial nerve canal presented a short way, and it reached the geniculate ganglion splitting in two branches (Figure 4B–D). The ventral branch ran through the canalis pro ramo nervi vidiani (Figure 4C,D) and reached the foramen nervi facialis, which is confluent with the foramen canalis carotici interni (Figure 1D–F and Figure 2E). This nerve continued anteriorly through the foramen nervi vidiani and pierced the pterygoid by the canalis nervus vidianus (Figures 1F, 2E and 4B–D). The canalis nervus vidianus bifurcates anteriorly, and it almost reached the foramen palatinum posterius. The hyomandibular branch of the facial nerve continued posterolaterally from the geniculate ganglion and runs ventral to the canalis cavernosus (Figure 4B–D), but without entering it. The distalmost end of this nerve exited the skull crossing the quadrate posteriorly to the mandibular condyle (Figure 1F). The glossopharyngeal nerve (cranial nerve IX) emerged from the cranial cavity posterior to the cavum labyrinthicum, and its canal crosses the processus interfenestralis of the opisthotic to reach the foramen externum nervi glossopharyngei (Figures 3D and 4C,D). The vagus and accessory nerves (cranial nerves X and XI) exited through the foramen jugulare anterius, a large foramen formed by the opisthotic and the exoccipital (Figure 4A–D). They entered the recessus scalae tympani, and both nerves left the skull through the foramen jugulare posterius, located in the exoccipital (Figure 2H). The hypoglossal nerve (cranial nerve XII) pierced the exoccipital across two posterolaterally directed canals (Figures 2H and 4A–D). The canalis alveolaris superior go across the maxilla in a trajectory parallel to the lateral edge of the bone (Figure 1C,F,H and Figure 4A,B). The posterior foramen of the canal is formed by the jugal and the maxilla. The canal reaches the nasal cavity anteriorly through the foramen alveolare superius.

The nasal cavity of *Euraxemys essweini* is identified as relatively small (Figure 4A–C), considering that it represents less than 20% of the total volume of the cranial cavity (including the nasal area). The olfactory region is weakly expanded dorsally (Figure 4C). The vestibular region is narrower than the lateral expanded anterior area of the nasopharyngeal ducts (Figure 4A,B). The ducts are directed posteriorly, defining straight trajectories (Figure 4B). The lateral edges of the ducts are almost parallel between them.

The cavum labyrinthicum is relatively low (Figure 4E,H), and its maximum height represents 77% of the medulla oblongata height (Figure 4C). The inner ear presents a ventrally weak-expanded vestibulum and elongated semicircular canals (Figure 4F,H). The anterior semicircular canal is formed on the prootic and the supraoccipital (Figures 2F and 3D). It is the longest canal of the system (Figure 4F,H). The posterior semicircular canal is defined by the opisthotic and the supraoccipital (Figure 2F) and the lateral by the prootic and the opisthotic (Figure 1C). The widest canal is the lateral one, which is dorsoventrally compressed, showing an oval cross-section (Figure 4F,H). The highest canal is the anterior one, whose dorsal surface represents the highest level of the labyrinth. The angle between both vertical semicircular canals (i.e., the anterior and the posterior) is recognized by 84° for the left inner ear and about 92° for the right one (Figure 4E,G). The crus communis

is almost longer than higher. Its dorsal surface is located below the dorsal level of the vertical canals (Figure 4F,H). The spaces defined by the vertical canals and the vestibulum are anteroposteriorly elongated and dorsoventrally narrow. The anterior space of the inner ears is longer and goes down more than the posterior.

The carotid artery entered the skull through the foramen posterius canalis carotici interni (Figure 1D–F, Figures 2E and 4B), defined by both the basisphenoid and the pterygoid (Figures 1D and 2E). The canalis caroticus basisphenoidalis crosses the basisphenoid anteromedially and reaches the pituitary fossa through the foramen anterius canalis carotici basisphenoidalis (Figure 2D). The angle formed by both carotid canals is acute, being almost 82° (Figure 4B).

## 4. Discussion

The analysis of the files obtained by the use of the computed tomography scan performed for the skull of the holotype and as so far only known specimen of *Euraxemys essweini* allow us to characterize some unknown osseous characters. Thus, considering the data matrix of the paper where it was defined (see [14]), three characters can be encoded for the first time. The foramen caroticum laterale of the pterygoid is recognized as absent (Character 76, stage 0); the clinoid process of the basisphenoid is dorsally expanded, and the basisphenoid presents a well-developed abducens canal (Character 107, stage 0); and the sella turcica is deep, and the margins of the dorsum sellae are well-defined (Character 110, stage 0). The last two characters could not be codified for Euraxemydidae by Gaffney et al. [14] since they were also unknown to the other representative of this lineage, i.e., *Dirqadim schaefferi*. However, the absence of the foramen caroticum laterale was subsequently recognized for *Dirqadim schaefferi* (see Character 97 in the supplementary data matrix of [27]), that state being recognized here as shared with *Euraxemys essweini*. The foramen caroticum laterale is present in Podocnemididae and most members of Chelidae, but its absence in Euraxemydidae is shared with the members of Pelomedusidae and Bothremydidae, as well as with some chelids such as *Chelus fimbriata* (Schneider, 1783) [14,27,38]. Within Pleurodira, the absence of a clinoid process and the presence of a groove for the abducens nerve instead of a developed canal are exclusively known for some representatives of Bothremydidae: the members of *Bothremys* and *Chedighaii* (Bothremydini) [14]. The presence of a very shallow sella turcica with low margins is an exclusive character of the genus of Bothremydidae *Taphrosphys* (Taphrosphyini) [14].

Although the cranial cavity in Testudinata is not very indicative of the shape or volume of the soft nervous tissues contained on it (e.g., [25,33]), differences can be observed between the diverse lineages of turtles (e.g., [21,29]). The angle formed between the forebrain and the hindbrain of the cranial cavity of *Euraxemys essweini* presents a relatively low value when it is compared with that in the members of Pelomedusidae and Chelidae, which present straighter cranial cavities with angles above 170° [29]. However, the angle of the cranial cavity measured for some Podocnemidoidea representatives (i.e., podocnemidids and bothremydids; see [29,30]) is lower than that observed for *Euraxemys essweini*. The lateral expansion of the cerebral hemispheres can be estimated using the measurements of different structures in the cranial cavity. In the case of *Euraxemys essweini*, both the ratio between the maximum width of the cranial cavity and its length (see supplementary material in [29]) and that between the width of the hemispheres and the medulla oblongata (see supplementary material in [27]) were measured. The first ratio in *Euraxemys essweini* (with a value of 0.34) represents a value higher than that in the chelids (with a value range between 0.20 and 0.26; see [29]). The value in *Euraxemys essweini* is close to that recognized for the pelomedusids (i.e., a value of 0.33 for *Pelusios niger* (Duméril and Bibron, 1835) [29,39]) but higher than that in the podocnemidid turtles [29]. In the case of the bothremydids, the ratio varies from values lesser than 0.30 for the members of Cearachelyini to the larger values recognized in some members of Taphrosphyini, reaching a value of 0.63 [30]. Considering the ratio between the width of the cerebral hemispheres and that of the medulla oblongata, some authors [27] proposed a differentiation between taxa with laterally slightly expanded

hemispheres (ratio less than 2.25) and others showing highly expanded ones (greater than 2.25). These character stages were reflected it in a data matrix (see Character 256 in the supplementary material of [27]). The value obtained here for *Euraxemys essweini* (1.66), being less than 2.25 as occurs with that in *Dirqadim schaefferi*, reflects a smaller expansion of the hemispheres than in most representatives of Chelidae or Bothremydidae. However, noticeable differences are identified within other groups of Pleurodira (i.e., Pelomedusidae and Podocnemidoidae), in which the values of these stages of character vary between forms with relatively little to highly expanded hemispheres (supplementary material in [27,29]). The cartilaginous ridge is recognized as very variable in shape and size within Pleurodira [29,40]. The presence of a weakly dorsally expanded ridge in *Euraxemys essweini* is shared with the members of Chelidae and Pelomedusidae [29]. The development of this structure is variable in Podocnemidoidea, showing a variable morphology, including the absence of this structure in some representatives such as, among others, *Erymnochelys madagascariensis* (Grandidier, 1867 [41]) [40]. The members of Bothremydidae show well-expanded cartilaginous ridges which, together with a relatively large lateral expansion of the cerebral hemispheres, represent a combination of characters unique to the lineage [29,30]. The development of this structure in Podocnemididae and closely related forms shows a high variability from forms in which it is absent (as in *Yuraramirim montealtensis* Ferreira, Iori, Hermanson, and Langer, 2018 [12] or *Amabilis uchoensis* Hermanson, Iori, Evers, Langer, and Ferreira, 2020 [27]) to representatives in which it is anteroposteriorly and dorsally well-developed (some *Podocnemis unifilis* Troschel, 1848 [42] specimens [29]).

Variations are recognized for the facial nerve system of Pleurodira, considering both the presence or absence of a canalis pro ramo nervi vidiani and the contact between the vidian branch and the geniculate ganglion with the canalis caroticus internus [28–30,35]. The geniculate ganglion of all pleurodires is housed in the prootic between the fossa acustico-facialis and the canalis cavernosus, but the ganglion clearly contacts the canalis caroticus internus exclusively in Chelidae and Pelomedusidae [29,35]. However, the geniculate ganglion lies very close to the canalis caroticus internus in Sahonachelyidae, nearly contacting it [28]. As occurs in *Euraxemys essweini* and *Dirqadim schaefferi*, the podocnemidids and bothremydids have a geniculate ganglion separate from the carotid canal [29,30,35]. The canalis pro ramo nervi vidiani is exclusive for the representatives of Podocnemidoidea within Pleurodira. This canal flows into the cavum pterygoidei in the podocnemidids, but it contacts the canalis caroticus internus in bothremydids. The condition observed here for *Euraxemys essweini* (unknown for *Dirqadim schaefferi*) is the same as that recognized for Bothremydidae. The anterior vidian nerve of *Euraxemys essweini*, anteriorly generating two branches, differs from that of *Dirqadim schaefferi*, which shows a single branch. The condition observed for *Euraxemys essweini* is shared with *Sahonachelys mailakavava* [28], not having been recognized for any other member of the Pleurodira [29,35].

The volume of the nasal cavity relative to the total volume of the cranial cavity in *Euraxemys essweini*, less than 20%, is shared with chelids and pelomedusids. This value is higher in podocnemidids and bothremydids (so far only having been obtained for freshwater representatives), reaching up to 28% in *Galianemys emringeri* [30], but it does not exceed 30% of the total volume. All these values obtained for the pleurodires are compatible with those observed in freshwater cryptodiran turtles, this condition having been related to a form of freshwater life in which the functions involving the nasal cavity (mainly the olfactory sense) would not be as developed as in terrestrial or pelagic sea turtles [19,21].

The inner ear in turtles is very conservative, with a poorly ventrally expanded vestibulum and low semicircular canals [18,32]. However, pleurodiran turtles show vertical semicircular canals (i.e., anterior and posterior canals) which are more anteroposteriorly elongated and thinner than in terrestrial or pelagic marine cryptodires [20,21,25,43]. All inner ears in Pleurodira show an M-shaped dorsal surface because the upper level of the crus communis lies below the apex of the semicircular canals (e.g., [21,28–30]). As in *Euraxemys essweini*, most chelids, pelomedusids, sahonachelids, and podocnemidids have an anterior semicircular canal higher than the posterior one. By contrast, the basal

Podocnemidoidae *Amabilis uchoensis* and most of bothremydids have more symmetrical semicircular canals in which the dorsal surfaces of both canals are approximately at the same level [27,30]. The shapes of the spaces formed between the vertical semicircular canals and the vestibulum are elongated in *Euraxemys essweini*, as in chelids, sahonachelyids, and podocnemidids [27,28]. By contrast, the ears of pelomedusids and bothremydids present spaces with an oval to circular shape [27,29,30]. In Chelidae, a range of variation between 80° (measured in *Chelus fimbriata* [29]) and 98° (measured in *Chelodina reimanni* Philipsen and Grossmann, 1990 [21,44]) has been documented for that angle formed between the vertical canals, the values obtained here for *Euraxemys essweini* being included in that range. The angle measured for representatives of Podocnemidoidea is also similar to those of the chelids, varying from 80° in *Podocnemis unifilis* [29] to 91° in *Galianemys whitei* [30]. The value observed in *Sahonachelys mailakavava*, being about 90° [28], is also compatible with those of the taxon studied here. The values of the angle for Pelomedusidae are less than 85°.

In *Euraxemys essweini*, as with most pleurodires, the carotid canals enter the skull from the foramen posterius canalis carotici interni and reach the posterior region of the pituitary fossa [45]. This condition is not shared with the members of Podocnemidoidae, in which the arteries enter the skull directly to the cavum pterygoidei [12,27,35]. In the representatives of this group of pleurodires, the carotids are only enclosed by bone in the basisphenoid portion [35]. Variation in relation to the bones that form the foramen posterius canalis carotici interni is identified within several pleurodiran clades (e.g., [14,28,35]). Within euraxemydids, while the foramen is exclusively formed by the pterygoid and basisphenoid in *Euraxemys essweini*, the prootic also participates in *Dirqadim schaefferi* [14]. The variability in the formation of this foramen in chelids and bothremydids is very high [14,35]. However, the bones that participate in the foramen in pelomedusids are the prootic together with the participation of the basisphenoid or the quadrate [29,35]. The angle formed by the carotid canals of *Euraxemys essweini* (82°) is much more acute than that in *Dirqadim schaefferi* (being about 114°, based on the three-dimensional reconstruction in the supplementary data of [27]). The angle formed by the carotid canals in chelids is closer to that of *Dirqadim schaefferi* than to *Euraxemys essweini* [29], while that of pelomedusids has been recognized as intermediate (being close to 95°; see [29]). The values for the angle between the carotid canals known for the podocnemidids are closer to that of *Euraxemys essweini*, this condition being also close to that in the Cearachelyini bothremydids [27,29,30]. The value of the angle is lower in *Euraxemys essweini* than in the other lineages of bothremydids, in which values of up to 140° have been documented [27,29].

## 5. Conclusions

The virtual three-dimensional reconstruction of the skull and neuroanatomical structures of the holotype and only known specimen of the Brazilian Albian *Euraxemys essweini* (Pleurodira, Euraxemydidae) is carried out here for first time. In addition, the processing of the files obtained by computed tomography scanning of the specimen have allowed us to observe some characters that could not be seen without the application of this non-destructive technology, increasing the anatomical information of this taxon. Part of the columella auris, unknown until now for *Euraxemys essweini*, is also described for the first time.

Neuroanatomical information within Pleurodira has been increasing in recent years with numerous publications, mainly on the neuroanatomy of Podocnemidoidea (i.e., the clade grouping Bothremydidae and Podocnemididae). However, neuroanatomical knowledge of other extinct lineages within the crown Pleurodira is still very poor. The comparison of the neuroanatomy of those extinct taxa with that of extant members of Pleurodira is a very useful tool that can allow us to determine the relationship between some features observed in the internal structures with taxonomic or environmental signals. In this sense, previous knowledge about some neuroanatomical elements of the Moroccan Cenomanian euraxemydid *Dirqadim schaefferi*, as well as the analysis presented here about *Euraxemys essweini*, have allowed us to significantly increase the information on the neuroanatomy of Euraxemydidae. The new data have been compared with the neuroanatomical characters

available for the other pleurodiran lineages in which their neuroanatomy had been studied. Several neuroanatomical characters recognized here for the cranial cavity of *Euraxemys essweini*, such as the possession of a cranial cavity with an angle of less than 170° or a greater lateral expansion of the lateral hemispheres, differ from those in Chelidae, whose representatives show a straighter cranial cavity and relatively less expanded hemispheres. The cranial cavity of *Euraxemys essweini* differs from those of the bothremydids in the absence of a dorsally expanded cartilaginous ridge. The lateral expansion of the hemispheres in *Euraxemys essweini* represents an intermediate condition between the less developed hemispheres in Podocnemididae and the wider ones of the Bothremydini and Taphrosphyini bothremydids. All these character states observed for the cranial cavity of *Euraxemys essweini* coincide with those in Pelomedusidae. The canal system that forms the facial nerve (which is not completely known for *Dirqadim schaefferi*) in *Euraxemys essweini* possesses a ventral canalis pro ramo nervi vidiani, and the geniculate ganglion does not contact the canalis caroticus internus. This canal system differs from that of Chelidae and Pelomedusidae, in which the geniculate ganglion directly contacts the canalis caroticus, so the canalis pro ramo nervi vidiani does not exist. In the case of Sahonachelyidae, although the facial nerve does not cross the canalis pro ramo nervi vidiani, it shares with *Euraxemys essweini* (but not with *Dirqadim schaefferi*) the bifurcation in the anterior region of the canalis nervus vidianus. The presence of the canalis pro ramo nervi vidiani in *Euraxemys essweini*, which is inferred for *Dirqadim schaefferi*, is also observed in the representatives of Podocnemidoidea, being an exclusive feature for these groups of Pleurodira. The shape of the inner ears in Pleurodira is very conservative throughout its evolutionary history. However, the greater length of the anterior semicircular canals relative to the posterior ones, along with the elongated shapes of the spaces formed between the vertical canals and the vestibulum observed for *Euraxemys essweini*, are more similar to those in Chelidae, Sahonachelyidae, and Podocnemididae than to those in Pelomedusidae and Bothremydidae. Within Euraxemydidae, *Euraxemys essweini* presents an acute angle between the canalis caroticus internus, similar to that in Podocnemididae and in the Cearachelyini bothremydids, while *Dirqadim schaefferi* shows greater similarities for this character with the chelids and some forms of Bothremydini.

**Author Contributions:** Conceptualization, M.M.-J. and A.P.-G.; methodology, M.M.-J. and A.P.-G.; software, M.M.-J.; validation, M.M.-J. and A.P.-G.; formal analysis, M.M.-J. and A.P.-G.; investigation, M.M.-J. and A.P.-G.; resources, A.P.-G.; data curation, M.M.-J. and A.P.-G.; writing—original draft preparation, M.M.-J.; writing—review and editing, M.M.-J. and A.P.-G.; visualization, M.M.-J.; supervision, A.P.-G.; project administration, A.P.-G.; funding acquisition, A.P.-G. All authors have read and agreed to the published version of the manuscript.

**Funding:** The research activity of the authors is funded by the Ministerio de Ciencia e Innovación (research project PID2019-111488RB-I00).

**Institutional Review Board Statement:** Not applicable.

**Data Availability Statement:** The skull of the holotype of *Euraxemys essweini* is deposited in the Palynology and Microvertebrata of the Paleozoic Collection of the Senckenberg Forschungsinstitut und Naturmuseum (Frankfurt, Germany).

**Acknowledgments:** The authors thank R. Brocke and L. Kraus for access to the *Euraxemys essweini* holotype and files related to it and thank S. Tränkner for the photographs of the specimen (Senckenberg Forschungsinstitut und Naturmuseum, Frankfurt) and Renate Rabenstein (Senckenberg CT-Lab Frankfurt) for the computed tomography scanning. The authors also thank the editor H. Huang and three anonymous reviewers for their comments and suggestions for the improvement of this manuscript.

**Conflicts of Interest:** The authors declare no conflict of interest.

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
