# Peer review of "Neuroanatomical Study and Three-Dimensional Cranial Reconstruction of the Brazilian Albian Pleurodiran Turtle Euraxemys essweini"

_diversity, doi:10.3390/d15030374_

Round 1

Reviewer 1 Report

The manuscript is nicely done and written.The study design is appropriate and apparently, the analyses were carefully performed.  I believe that the results are valuable for the scientific community and has significant scientific merit, as it will probably ignite many further studies in the near future. The literature is up to date and conclusions are justified by obtained results. I do not have any emerging concerns about the scientific aspects of the manuscript. However, the manuscript could be improved by making some changes as suggested as follows:

Line 63 – The aim of the study is not known. Please provide any goal or hypothesis.

Line 66-69 – any abbreviations should be listed in a separate chapter. Not as a part of introduction.

Line 69 and throughout the text - All Latin names should be written in italics. Sometimes it is annoying that authors mix English and Latin names (carotid vs. caroticus; facial nerve vs. nervi facialis; canal vs. canalis, etc.). Please unify it.

Line 93 – please provide details of hardware producer (ProCon-X-Ray-Micro-CT scanner).

Line 107 – Please provide any details for Avizo 7,  Geomagic Studio 2014.3.0 as well as Adobe Photoshop software/tools. Add license if needed.

Line 223 – “relatively small” - In relation to what? Please explain.

Line 386 – Conclusions at the present form are too long. Please make them more concise.

Figure 4 -It is unclear what is scale bar for C-D?

Author Response

Reviewer 1 exposed general comments:

  1. “Line 63 – The aim of the study is not known. Please provide any goal or hypothesis”. Response: This has been done.

  1. “Line 66-69 – any abbreviations should be listed in a separate chapter. Not as a part of introduction”. Response: As in other manuscripts published in Diversity (including some of ours), the abbreviations are included in a subsection at the end of the section Introduction.

  1. “Line 69 and throughout the text - All Latin names should be written in italics. Sometimes it is annoying that authors mix English and Latin names. Please unify it”. Response: The use of italics for neuroanatomical structures is not common in works on neuroanatomy, so it has not been used in this work. In the same way, in numerous publications both Latin and English are used interchangeably to name these structures. See for example Gaffney et al. (2006) or Rollot et al. (2021). In this sense, Rabi et al. (2013) indicated that, for these studies, "italics added for emphasis".

  1. “Line 93 – please provide details of hardware producer (ProCon-X-Ray-Micro-CT scanner)”. “Line 107 – Please provide any details for Avizo 7, GeomagicStudio 2014.3.0 as well as Adobe Photoshop software/tools. Add license if needed”. Response: All the information provided for the methodology in the present manuscript (the scanner model, the scanning parameters, the software used...), and included in the section Material and methods, is that commonly provided in this type of studies (as can be verified by looking at any of the papers cited in the references section).

  1. “Line 223 – “relatively small” - In relation to what? Please explain”. Response: The size of the nasal cavity is relatively small considering the total volume of the cranial cavity, as is now better explained in the text.

  1. “Line 386 – Conclusions at the present form are too long. Please make them more concise”. Response: This has been done.

  1. “Figure 4 -It is unclear what is scale bar for C-D?”. Response: Thanks for reporting this error. The figure has been improved including the scale of 4C and 4D.

Reviewer 2 Report

Dear Authors,

Find attached some minor style and spelling things to fix. 

Sincerely, 

Author Response

Reviewer 2 exposed general comments:

  1. “Double check this sentence. Euraxemys is part of Euraxemydidae and not its sister taxon as it is written here”. Response: The sentence was revised and the expression sister taxon was deleted.

  1. “Just double check if the journal obligates to referer the figures inside the manuscript in order Fig. 2A, Fig. 2B before this one”. Response: This has been corrected.

  1. “Merge the references (40, 41)”. Response: The reference (41) corresponds to the authorship of the species Erymnochelys madagascariensis. This reference is now included inside the parentheses.

  1. “-”in the institutional acronym. Response: This symbol is not including in the acronym of this institution.

In addition, minor changes were proposed by the Reviewer 2 that had been considered by the authors.

Reviewer 3 Report

This is a solid contribution to the literature on both the early evolutionary history of pleurodires and reptilian neurosensory evolution.  It adds new  new morphological characters for a fossil pleurodiran that has a contentious phylogenetic placement, with some previous analyses supporting it as an early stem podocnemid while others support a stem pelomedusoid identity.

I have a few recommendations that should improve the manuscript:

· The phylogenetic analysis using the matrix from Gaffney et al. (2006) should be replicated after including the newly scored characters. The results will show the impact of these characters in interpreting the evolutionary relationships of Euraxemys essweini /Eurxemydidae

· Add measurements to support the claim that the shape of the pituitary fossa is slightly longer than wide (lines 136 and 137)

· On line 177, when explaining the lengths of axes of the pituitary fossa, it is unclear to what “the lateral one” is referring to, maybe replace that with mediolateral axis.

· On line 272 it is stated that the cranial cavity is not indicative of shape or volume of soft nervous tissue in reptiles, but at least in archosaurians there is a more clear correlation between these. Maybe changing this to refer only to turtles instead of reptiles would be a more accurate statement.

· Review sentence composition on lines 29, 63, 291, 302

Author Response

Reviewer 3 exposed seven general comments:

  1. “The phylogenetic analysis using the matrix from Gaffney et al. (2006) should be replicated after including the newly scored characters. The results will show the impact of these characters in interpreting the evolutionary relationships of Euraxemys essweini /Eurxemydidae”. Response: The authors do not intend to make a detailed anatomical study in this paper. The analysis of the cranial characters of Euraxemys essweini through the use of CT would be interesting in the future, also adding information relative to the review of the postcranial elements, to increase knowledge about this species and the precise systematic identification of its clade within Pleurodira.

  1. “Add measurements to support the claim that the shape of the pituitary fossa is slightly longer than wide (lines 136 and 137)”. Response: This has been done.

  1. “On line 177, when explaining the lengths of axes of the pituitary fossa, it is unclear to what “the lateral one” is referring to, maybe replace that with mediolateral axis”. Response: This has been done.

  1. “On line 272 it is stated that the cranial cavity is not indicative of shape or volume of soft nervous tissue in reptiles, but at least in archosaurians there is a more clear correlation between these. Maybe changing this to refer only to turtles instead of reptiles would be a more accurate statement”. Response: This has been done.

5. “Review sentence composition on lines 29, 63, 291, 302”. Response: This has been done.